

# Gridded Flood Depth Estimates from Satellite Derived Inundations

Seth Bryant[1,2], Heather McGrath[3], Mathieu Boudreault [4]

[1] GFZ German Research Centre for Geosciences, Section 4.4. Hydrology, Potsdam, Germany
[2] Institute for Environmental Sciences and Geography, University of Potsdam, Germany
[3] Canada Centre for Mapping and Earth Observation, Natural Resources Canada, Ottawa, Canada
[4] Department of Mathematics, Université du Québec à Montréal, Montréal, Canada

*Correspondence to*:  Seth Bryant (bryant.seth@gmail.com)

**Abstract.** Canada's RADARSAT missions improve the potential to study past flood events; however, existing tools to derive
flood depths from this remote-sensing data do not correct for errors, leading to poor estimates. To provide more accurate
gridded depth estimates of historical flooding, a new tool is proposed that integrates Height Above Nearest Drainage and Cost
Allocation algorithms. This tool is tested against two trusted, hydraulically derived, gridded depths of recent floods in Canada.
This validation shows the proposed tool outperforms existing tools and can provide more accurate estimates from minimal
data without the need for complex physics-based models or expert judgement. With improvements in remote-sensing data, the
tool proposed here can provide flood researchers and emergency managers accurate depths in near-real time.

## 1 Introduction

Flooding has become the costliest natural disaster in Canada, with economic losses estimated around $2.5 billion per year
(Office of the Parliamentary Budget Officer, 2016). To mitigate this flood risk, large investments in infrastructure and planning
have been made by the federal government in the past decade (National Disaster Mitigation Program, 2017; Government of
Canada, 2021); however, accuracy and the absence of data on historical flooding remains challenging for the models
underpinning these investments (McGrath et al., 2015; Bryant et al., 2021). While new satellite missions have improved
capabilities for mapping inundation extents, data on maximum flood depth, which is commonly found to be the most significant
indicator of building damage following European floods (Mohor et al., 2020; Laudan et al., 2017; Merz et al., 2010), remains
scarce. The absence of such depth data in Canada limits the utility of flooding research, ultimately leading to less informed
flood management decisions.

Relying on microwave pulses that can reflect the ground surface at night and through clouds, Synthetic Aperture Radar (SAR)
instruments have become an important tool for measuring flood inundation at large scales (Shen et al., 2019).  For example,
the recently launched, three-satellite, RADARSAT Constellation Mission provides regular medium-resolution (30-100 m)
SAR observations across Canada but can capture high-resolution (1-3 m) observations when requested for flood disasters
(Canadian Space Agency, 2021b). A common approach for identifying flooded areas from SAR observations employs a


threshold to the measured backscatter values to classify water covered areas based on their surface roughness (Benoudjit, 2019). For example, the 'Floods in Canada' (FiC) project calculates a backscatter threshold from historical inundation data to classify open-water flooding; before identifying adjacent areas with flooded vegetation using a second threshold (Natural Resources Canada, 2020).

While SAR measurements are advantageous for identifying inundated areas remotely (compared to optical measurements), the signal technology has limitations. Shen et al. (2019) identify three common error sources challenging SAR-derived inundation algorithms: 1) smooth dry-surfaces that return similar signals to inundated surfaces or rough water surfaces that return uncharacteristically rough signals; 2) georeferencing of SAR images (which often relies on ancillary terrain data); and 3) inundated areas near dense obstructions that scatter returning signals. These errors lead to less accurate inundation predictions

in areas with: dense urban infrastructure, dense vegetation, floating debris (e.g., ice), waves/rapids, steep river-banks perpendicular to the instrument (Natural Resources Canada, 2020; Cian et al., 2018), or recent construction/earthwork.

Simplified conceptual or '0D' inundation models provide an alternate means for estimating inundation efficiently; however, some calibration data is required to simulate specific events (Teng et al., 2017). Height Above Nearest Drainage (HAND) methods are a raster-based class of 0D models that leverage a digital elevation model (DEM) and drainage network information

to implement a three-phase routine for identifying flooded regions: 1) generate a hydraulically conditioned DEM; 2) calculate the height of each cell above the drainage network or 'HAND value'; then 3) map all cells below some HAND value threshold, typically derived from observations or rating curves (Rodda, 2005; Rennó et al., 2008; Donchyts et al., 2016). In the U.S., HAND techniques were coupled with the National Weather Model to produce uncalibrated continent-scale 10 m resolution inundation predictions (Liu et al., 2018) that were shown to accurately capture 19-25% of inundated areas (Johnson et al.,

50 2019).

Advancements in remote sensing and terrain analysis have improved the availability and accuracy of historical inundation data; however, the corresponding (higher-dimensional) gridded depth data, desired by flood vulnerability research, has proven more illusive. Cian et al. (2018) provides a review of methods to derive gridded depths from remote sensing data, starting with work that manually overlaid LANDSAT imagery on terrain contours to estimate reservoir volumes (Gupta and Banerji, 1985).

This class of 'inundation-polygon terrain-overlay' methods seeks to first construct a water surface by identifying and projecting the land-water interface or 'shoreline', before subtracting the DEM from this water surface to yield gridded depths. When such methods are employed with DEMs that omit or ignore bathymetry, depth values are underestimated for waterbodies. A second, more relevant, error source is introduced by inaccuracies in the inundation extents or polygons. In areas with significant topography (e.g., steep riverbanks) small errors in inundation polygons can yield large errors in shoreline elevations (Nguyen

et al., 2016).

Within the inundation-polygon terrain-overlay class of depth estimation methods, a distinction can be made between those methods assuming a flat-water shoreline (or constant elevation) and those that are agnostic or allow shoreline elevation


heterogeneity to propagate into the water surface estimates. Flat-water shoreline methods are well-suited to floods with near-zero water surface gradients and high segmentation between inundation polygons (Cian et al., 2018; Gupta and Banerji, 1985).

Shoreline agnostic methods are better suited to handle floods with higher water surface gradients and more continuous inundation polygons; however, these methods are more sensitive to errors in shoreline location (Cohen et al., 2018; Brown et al., 2016; Nguyen et al., 2016). A second distinction can be made between those requiring expert input for parameterization (Cian et al., 2018; Brown et al., 2016; Nguyen et al., 2016) and those that are automated or only require simple data inputs (Cohen et al., 2018).

Early attempts to estimate gridded-flood depths from SAR-derived inundations implemented largely manual workflows. For a 2014 flood of a low-lying area in the U.K., Brown et al. (2016) selected shoreline elevation points (from LiDAR) from which the flooded water surface was interpolated and finally subtracted from the elevation model to obtain gridded depths. This method was compared to LiDAR measurements of the flood surface, yielding a root-mean-square-error (RMSE) of 15 cm for overlapping inundation cells. Rather than use the SAR-derived inundations directly, Nguyen et al. (2016) used them to identify

a best-fitting parameterization of a simplified 2D hydrodynamic model for a low-lying floodplain in Vietnam. No RMSE was reported, and the method failed to predict inundation in "several small [sub] areas".

Cian et al. (2018) developed a semi-automated flat-water method employing statistical analysis of the raw shoreline elevations to identify the value where 5 percent of adjacent sorted values differ by less than 10 cm. Manual correction was used for inundation polygons whose raw shoreline elevation values failed to yield a conforming elevation. These results were compared

against hydrodynamic modelling derived depths, yielding a RMSE between 55 cm and 79 cm for overlapping inundation cells.

Leveraging user supplied flood path transect lines and boundary masks, Scorzini et al. (2018) developed the 'RAPIDE' tool. This shoreline agnostic tool was tested against a hydrodynamic simulation of a flood in Italy using the hydrodynamic inundation for the tool input (rather than satellite derived inundations), yielding a RMSE between 38 and 79 cm for overlapping inundation cells.

Using a fully automated open source algorithm, Cohen et al. (2018) developed the Flood Water Depth Estimation Tool (FwDET) version 1.0 for the proprietary ArcGIS platform using a raster-based shoreline agonistic 'nearest boundary cell elevation' routine to interpolate shoreline elevations onto the interior inundated region. Version 2.0 replaced the interpolation routine with a more efficient 'cost allocation' routine better suited for inundations with incomplete boundaries (e.g., coastal flooding) (Cohen et al., 2019). Cohen et al. (2019) tested this tool against hydrodynamic results for two flood-prone regions

in the U.S. using the hydrodynamic inundation for the tool input (rather than satellite derived inundations) and found errors exceeding 1.5 m. Version 2.0 was later ported to Google Earth Engine (Peter et al., 2020). Cohen et al. (2019) reports on a second tool 'FwDET-QGIS', similar to Version 1.0 but for the QGIS open source platform, with the GRASS 'r.grow.distance' function (GRASS GIS manual, 2021) providing the nearest boundary cell elevation routine (Raney and Cohen, 2019). With the exception of a two-fold decrease in run-time (compared to Version 2.0), no comparison was reported for FwDET-QGIS


(Cohen et al., 2019). Aside from a low-pass filter applied to depth results, the FwDET tools propagate all shoreline errors into depth estimates.

All the aforementioned depth estimation methods report accuracy by comparing against some trusted source for all overlapping grid cells, rather than comparing depth estimates at asset locations (e.g., buildings) — the metric generally sought by flood vulnerability research. Further, no study with a fully automated method reported accuracy against satellite derived inundations.

In this context, our study pursues the following objectives: 1) present a fully automated shoreline agnostic tool that provides moderate error correction of shorelines; and 2) test this tool and FwDET-QGIS on two recent floods in Canada using typical Canada-wide datasets.

## 2 Methods

This study develops the novel Rolling HAND Inundation Corrected Depth Estimator (RICorDE) Tool for generating gridded

depth estimates of past flood events from approximate inundation polygons and a DEM. This tool allows for moderately varying water-land interface 'shoreline' values and does not require expert input. To demonstrate the accuracy of RICorDE, depth estimates are generated for two historical flooding events in Canada using publicly available datasets. These satellite derived depth grids are then compared against 'trusted' depths simulated by others with more sophisticated site-calibrated hydraulic models. To provide a comparison, the FwDET-QGIS tool is also tested against the same datasets.

### 2.1 RICorDE v1

RICorDE produces gridded water depth estimates by incorporating a HAND sub-model and cost distancing algorithms to extrapolate edge values into the inundated region. Built for estimating depths from an approximate polygon produced by the 'Floods in Canada' (FiC) project, RICorDE draws all input data from Canadian web-hosted sources, only requiring the user to specify the period and area of interest. However, users can supply similarly formatted input data from alternate sources.

Following data downloading, input and pre-processing, RICorDE uses the WhiteboxTools 'ElevationAboveStream' tool (WhiteboxTools User Manual, 2021) to generate the HAND values raster from user-supplied permanent waterbody polygons and the DEM. From these inputs, the central depth estimating algorithm of RICorDE has three phases: 1) hydraulically correcting the approximate inundation polygon to remove egregious over-predictions; 2) interpolating rolling HAND values for the flooded domain and corresponding water surface levels (WSL); and finally; 3) subtracting the water level grid from the

DEM. The remainder of this section provides additional detail on these three phases of the algorithm.

The first phase to hydraulically correct the approximate raw inundation polygon is summarized in Fig. 1. To address under-predictions in the raw polygon (areas falsely shown as dry), user supplied polygons denoting permanent water bodies are used to fill erroneous areas (Fig. 1 Panel 1). To address over-predictions (areas falsely shown as wet) the newly corrected inundation polygon is used to generate shoreline HAND values along valid edges (Fig. 1 Panel 2 and 3). From these samples, the upper



or third quartile (value between the median and maximum) is calculated and used to generate a HAND inundation polygon by polygonizing a mask of all lesser HAND raster values (Fig. 1 Panel 4). Finally, this new 'maximum inundation' polygon is used to clip all external values from the approximate inundation (Fig. 1 Panel 5). In this way, egregious over- and under-predictions in the raw inundation polygon are corrected before proceeding with the algorithm.

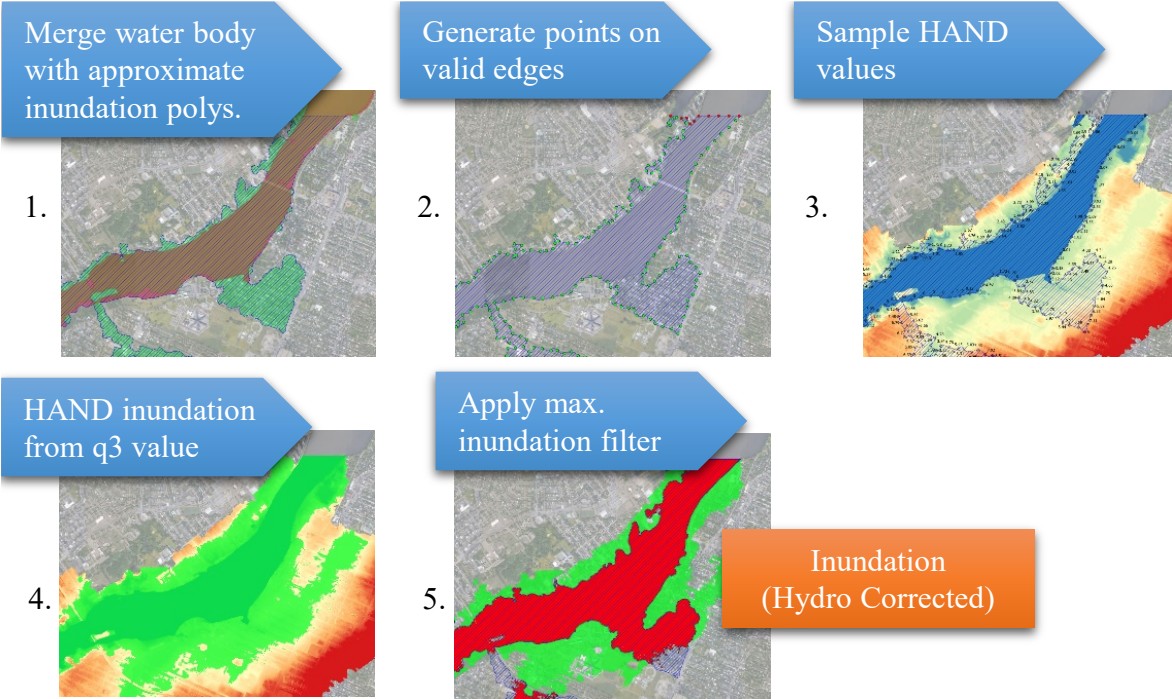

**Figure 1: Process diagram of the first phase of RICorDE's depth algorithm for generating the hydraulically corrected inundation polygon showing five basic steps; where 'q3' is the third quartile of the sampled HAND values. Basemap imagery from © Maxar Technologies.**

The second phase of RICorDE's depth algorithm is summarized in Fig. 2 and develops WSLs from the hydraulically corrected inundation of the previous phase. This phase first generates an interior surface of HAND values that best represents the flooding

in each grid cell ('Rolling HAND Grid') from which the final WSL is generated by mosaicking the HAND derived WSLs corresponding to each cell. This process begins with a second sampling of the shoreline HAND values using the hydraulically corrected inundation of the previous phase (Fig. 2 Panel 1). These second-generation values are again filtered using statistics from the initial first-generation HAND sampling: where a lower bound is forced from the first quartile and an upper bound is forced from the third quartile (Fig. 2 Panel 2). This second aspatial filtering of HAND values is required to address sampling

errors that arise from small spatial shifts introduced in the polygonization process of the hydraulic correction in the first phase (Fig. 2 Panel 4), while preserving the inundated area. From these corrected values, continuous shoreline HAND values are generated by first interpolating using an Inverse Distance Squared Weighting algorithm (GRASS GIS manual, 2021) as shown in Fig. 2 Panel 3, before masking out the interior region (Fig. 2 Panel 4). Interior HAND values are then generated using the





WhiteboxTools 'CostAllocation' algorithm (WhiteboxTools User Manual, 2021) as shown in Fig. 2 Panel 5. This algorithm
is similar to the one used by FwDET version 2.0 and internal testing showed this produced more hydraulically reasonable
results than the Inverse Distance Squared Weighting from previous steps. The final Rolling HAND Grid is obtained by
smoothing with a low-pass filter (GRASS GIS manual, 2021) iteration loop until a HAND value gradient less than or equal to
0.1 m/m is achieved (Fig. 2 Panel 6). With this smoothing, the algorithm balances the flooding surface implied by the
hydraulically corrected inundation and a flooding surface (of constant height) derived from the DEM. By smoothing HAND
values, rather than WSLs, the algorithm generates results with a bias towards the vertical profile of the flow path (e.g., WSL
in a flooded river channel), rather than a flat-water surface.

Once the 'Rolling HAND Grid' is obtained, the second phase continues by generating a HAND inundation raster for each
unique value within the rolling grid, following the same procedure described in phase 1 (Fig. 2 Panel 7). For each of these
inundations, a corresponding WSL raster is generated by first masking out interior regions on the DEM, then applying the
'CostAllocation' algorithm to grow the shoreline values into the interior, and finally masking out the exterior (Fig. 2 Panel 8).
This is similar to the FwDET version 2.0 routine; however, here more realistic inundation regions obtained from the HAND
model are used (rather than approximate inundation polygons). Because these inundations are derived from the DEM itself,
DEM artifacts do not lead to internal inconsistencies within the WSL rasters. Phase 2 concludes by mosaicking values from
the WSL set according to the positions of the corresponding HAND value found in the rolling HAND grid, then applying a
final low-pass filter (Fig. 2 Panel 9).

In the third and final phase of RICorDE's depth algorithm, the DEM is subtracted from the WSL Mosaic to obtain the raw
depth raster. From this raster, depths less than or equal to zero are removed, and the remaining raster is clipped to match the
extents of the hydraulically corrected inundation (from the first phase).


**Figure 2: Process diagram for the second phase of RICorDE's depth algorithm for generating the Rolling HAND Grid and the Water Surface Level (WSL) Mosaic. Basemap imagery from © Maxar Technologies.**

**2.2 Evaluation**

To evaluate the performance of RICorDE v1's novel depth estimation algorithm, approximate SAR-derived inundation polygons produced by the Floods in Canada project (FiC) for a 2018 flood in New Brunswick and a 2017 flood in Quebec, Canada are used to generate gridded depth estimates. Additional gridded depth estimates are generated using FwDET-QGIS (Cohen et al., 2019) for comparison. These simulated depth grids are then tested against corresponding 'trusted' grids obtained from hydraulic modelling done by others. To test performance, metrics based on the depth sampled at building locations are used.



### 2.2.1 Study Flood Events

Two recent spring fluvial flooding events in Canada were selected for evaluation.

*Rivière des Prairies at Montreal, Quebec — May 2017 Flood:* The Rivière des Prairies is a deltaic channel dividing Laval from Montréal, two cities in Southern Quebec at the confluence of the regulated St. Lawrence River and Ottawa River. The confluence has a drainage area of roughly 240,000 km$^2$ and 150,000 km$^2$ for the St. Lawrence and Ottawa rivers respectively. Above-average precipitation in April and early May 2017, combined with snowmelt, contributed to the highest flow on record

for the Ottawa River and regulators discharging the maximum allowable flow to the St. Lawrence River (Teufel et al., 2019). On May 8th, 2017 the water level peaked near the mouth of the Rivière des Prairies at an elevation of 24.8 m, the highest in the 26-year record, while the discharge peaked at 3,310 m$^3$/sec (Environment and Climate Change Canada Historical Hydrometric Data, 2021). Following the 2017 event, the Communauté métropolitaine de Montréal (CMM) calibrated a 2D hydrodynamic model of the Rivière des Prairies using high water measurements obtained during the 2019 peak, which reached

levels within 5 cm of the 2017 peak. Using this model, a map estimating the 2017 WSLs was produced by CMM. However, levees and temporary flood mitigation structures were not included in the model (i.e., some areas that remained dry in 2017 are shown as flooded on CMM's map). To correct for this, levee-protected areas identified through visual inspection as shown in Fig. S1 were removed for this analysis.

*Saint John River at Fredericton, New Brunswick — May 2018 Flood:* The Saint John River is a regulated river that drains

roughly 55,000 km$^2$ of mostly forested regions within Maine (U.S.), Quebec (Canada), and New Brunswick (Canada) before reaching the Bay of Fundy at Saint John in New Brunswick (Newton and Burrell, 2016). The lower broad and shallow reaches of the river form the flood-prone New Brunswick Lowlands, home to the provincial capital of Fredericton and numerous recorded flood disasters, most notably in 1973, 2008 (McGrath et al., 2015), and 2018 (Hrabluk, 2019). Triggered by rapid snowmelt and rain, on May 1st, 2018 the Saint John River discharge peaked at 6,070 m$^3$/sec and WSL at 8.2 m at Grand Falls

and Fredericton respectively, the fourth highest WSL in the 85-year record (Environment and Climate Change Canada Historical Hydrometric Data, 2021). An estimated 12,947 buildings were damaged by the flood and $CAD 80 million in government relief was estimated (Hrabluk, 2019).

Following the 2018 event, the Province of New Brunswick (PoNB) generated a maximum WSL map through a combination of hydraulic modelling and analysis of the six water level gauges (Environment and Local Government, 2021). This result was

validated against aerial imagery of the 2018 peak inundation and high-water marks surveyed during the flood peak of the following year, which had a gauge reading at Fredericton within 5 cm of the 2018 level (Environment and Climate Change Canada Historical Hydrometric Data, 2021).





### 2.2.2 Tool Inputs

Three publicly available datasets produced by Natural Resources Canada provided the primary inputs for the depth estimation
tools used in this evaluation: 1) DEMs were sourced from the High-Resolution Digital Elevation Model (HRDEM); 2)
approximate inundation polygons were sourced from the Floods in Canada project (FiC); and 3) permanent waterbody
polygons were obtained from the National Hydro Network project (NHN). The remainder of this section summarizes these
initiatives.

Since 2011, Natural Resources Canada has been modernizing Canada's national terrain model. This has involved the collection
of LiDAR derived DEMs south of the productive forest line, and a DEM based on autocorrelation of high-resolution optical
images among other remote sensing methods (e.g., radar interferometry), with a basic resolution of 1-2 m (Government of
Canada and Natural Resources Canada, 2020). As of July 2021, HRDEM covers nearly 500,000 km$^2$ of Canada, including 61
of the largest cities and the two study areas.

NHN is a national database of inland waters, providing vector data of waterbodies, watercourses, reservoirs, man-made
obstructions (e.g. dams), etc. from the best available provincial and federal collections of data, at 1:50,000 scale or better
(Government of Canada, 2004).

In December 2007, the Canadian Space Agency launched the RADARSAT-2 mission which carries a 15 m Synthetic Aperture
Radar (SAR) antenna (Canadian Space Agency, 2021a). Expanding disaster monitoring capabilities, the RADARSAT
Constellation Mission was later launched in June 2019 (Canadian Space Agency, 2021b). A primary user of these missions is
the Emergency Geomatics Services team at Natural Resources Canada. During major flood events, both optical and SAR
satellite imagery are used in conjunction with ancillary data layers to generate near-real time mapping of flood events to support
emergency response activities as part of the FiC project (Natural Resources Canada, 2020). A multi-step process is employed
to map open water and flooded vegetation, which includes supervised machine learning classification and threshold-based
region growing. The FiC data repository (https://open.canada.ca/data/en/dataset/08b810c2-7c81-40f1-adb1-c32c8a2c9f50)
contains active flood extents (current to past 72 hours) and archived extents dating back to 2011.

For the Saint John River event, the FiC scene from May 3$^{rd}$, 2018 was selected. For the Rivière des Prairies event, the FiC
scene from May 9$^{th}$, 2017 was selected. Corresponding metadata reports both scenes have a 'moderate' confidence level and
are SAR derived. Maps summarizing the tool data inputs used for both events are provided in Fig. S1 and Fig. S2.

### 2.2.3 Performance Metrics

To test the performance of the depth-estimate tools, similar studies often compare per-cell depth values (Nguyen et al., 2016;
Brown et al., 2016; Cian et al., 2018; Cohen et al., 2018; Scorzini et al., 2018), biasing the performance towards areas without
assets (i.e., focusing on the whole domain rather than building locations). In regions with heterogeneous asset densities, like
in the two areas of this study where development has occurred along riverbanks, this per-cell performance reporting strategy



is less useful for flood vulnerability researchers who are interested in the exposure of assets (e.g., buildings) not open
floodplains. Further, this per-cell reporting often obscures the performance of binary wet/dry predictions by only calculating
metrics for overlapping cells (i.e., where both the validation and estimated grids indicate flooding).

To address these challenges, this study focuses on metrics based on building locations and reports performance for inundation
(wet vs. dry) predictions separate from depth value predictions. This separation allows for a more robust evaluation of each
tool as inundation predictions are more sensitive to the input inundation polygon, while the depth value predictions are more
closely related to algorithm performance.

For both inundation and depth metrics, depth values are first sampled from the grids/rasters at asset locations obtained from
centroids of the 'CanadianBuildingFootprints' projects (Microsoft, 2019). For each tool, the inundation performance is then
calculated against the trusted grid. To report on the performance of the depth predictions, samples with zero value are
discarded. To calculate difference and correlation metrics, first the trusted and simulated depth sets (with zeros removed) are
paired. For the correlation analysis, missing pair values are discarded. For the difference analysis, missing pair values are
replaced with zeros before subtracting each trusted depth from its paired simulated depth. Because each depth raster differs in
extents (i.e., which cells are predicted wet/dry), this paired analysis results in different size datasets within the same trial for
each tool comparison.

## 3 Results and Discussion

Raw asset samples for the gridded-depth simulations generated by RICorDE and FwDET-QGIS, and corresponding trusted
values, for the two flood events are summarized in Table 1. This table also summarizes three performance metrics by
comparing simulated values against the trusted values at each asset: 1) root mean square error (RMSE); 2) mean of all
difference values (simulated – trusted); and 3) the correlation coefficient. This table shows that RICorDE outperforms FwDET-
QGIS for five of the six metrics and trials considered. The remaining underperforming metric, the 'difference mean' of the
Saint John River event, is discussed below.





**Table 1: Summary of the trusted and simulated depth grids for the two study floods.**

| Depth Grid Code | Study Flood Event Trial | Method | Depth Samples | | | | |
| --- | --- | --- | --- | --- | --- | --- | --- |
| | | | Raw Values | | Performance | | |
| | | | count | max | RMSE | difference mean | r value |
| SJ_PoNB | 2018 Saint John River | trusted | 1045 | 6.0 | | | |
| SJ_FQ | 2018 Saint John River | FwDET-QGIS | 1108 | 29.6 | 1.942 | 0.119 | 0.202 |
| SJ_RIC | 2018 Saint John River | RICorDE | 569 | 6.6 | 0.790 | -0.306 | 0.582 |
| dP_CMM | 2017 Rivière Des Prairies | trusted | 542 | 1.3 | | | |
| dP_FQ | 2017 Rivière Des Prairies | FwDET-QGIS | 1741 | 12.2 | 0.606 | 0.155 | 0.118 |
| dP_RIC | 2017 Rivière Des Prairies | RICorDE | 776 | 3.1 | 0.509 | 0.101 | 0.147 |

To show the accuracy of inundation predictions (i.e., wet vs. dry) from the two tools using the FiC approximate polygons, Fig.
3 shows the portion of over- (tool predicts the asset is wet in error), under- (tool predicts the asset is dry in error), and accurate-
260 inundation predictions. This figure shows that both tools yielded less accurate *inundation* predictions for the Des Prairies trial
(than for the Saint John), in contrast to the more accurate *depth* value predictions shown for this trial in Table 1.

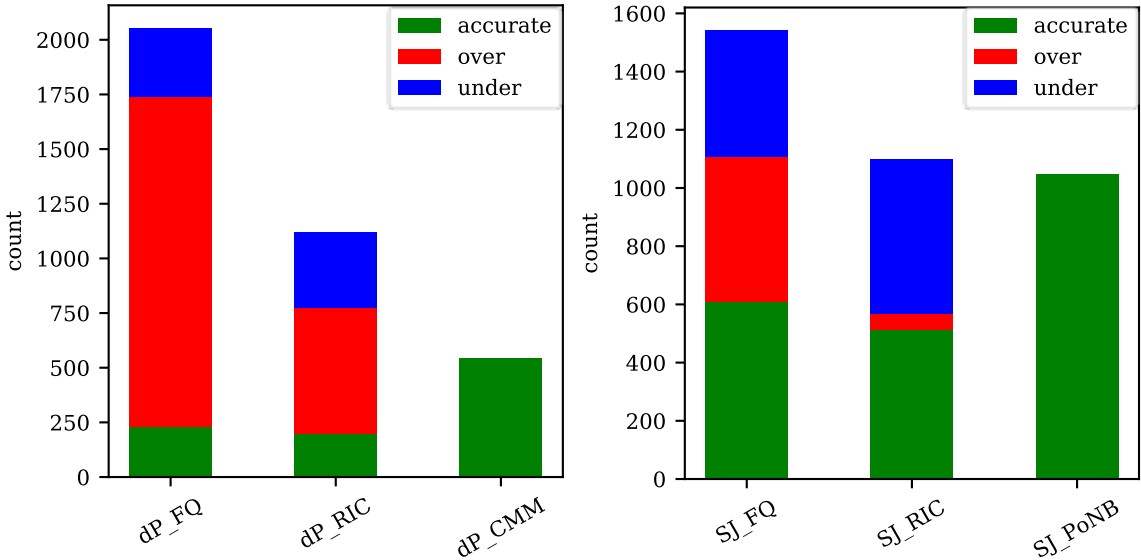

**Figure 3: Wet/dry asset sample performance for the two study floods showing: [left] 2017 Rivière Des Prairies performance and [right] 2018 Saint John River performance. See Table 1 for additional legend.**

265 Because FwDET-QGIS maps onto the raw inundation approximation (i.e., no hydraulic correction), values for these
simulations ('FQ') reflect the accuracy of the FiC polygons themselves, rather than some underlying algorithm. From this, Fig.





3 shows that the two FiC scenes investigated both over- and under-predict inundation of assets; however, over-prediction is more severe. This is especially true for the Rivière des Prairies event (Fig. 3 'dP_FQ'), where the FiC polygons predicted less than half of the total assets accurately (228 of 542). This could be attributed to erroneous treatment of levees in the trusted data 270  'dP_CMM' (rather than errors in the FiC polygons); however, comparing the two trials suggests this possible artifact is less relevant than the inaccuracies inherent in the FiC polygons. Alternatively, the performance improvement between the 2017 des Prairies and the 2018 Saint John trials could also be attributed to advancements in the FiC project made during the year between the two floods.

From Fig. 3, the advantage of RICorDE's hydraulically corrected inundation can be seen in the relative decrease of 'over' 275  predictions (compared to FwDET-QGIS). However, RICorDE performs slightly less well with 'under' prediction errors (predicts dry when the trusted shows wet). This 'dry bias' could be attributed to some over-correction of the inundation during the second phase of the depth algorithm (Fig. 2 Panel 2). This bias is also reflected in the lower (more negative) 'difference mean' values shown on Table 1 (relative to FwDET-QGIS), leading this metric to underperform for the Saint John Event despite the lower RMSE (0.657 vs. 1.915 m). If instead the *absolute* difference values are examined, the mean for RICorDE 280  would outperform by 23 cm (0.643 vs. 0.876 m). In other words, the combined effect of slightly better under predictions and much worse over predictions somewhat balance, pulling the difference mean of the FwDET-QGIS Saint John trial closer to zero, despite having more erroneous predictions.

To demonstrate the performance of the simulated depth values for the 2018 Saint John River trial, sample values, difference values, and correlation plots are provided in Fig. 4.

**Figure 4: Performance metrics matrix plot with rows for: [3] RICorDE simulated, [2] FwDET simulated, and the [1] trusted depth grid of the 2018 Saint John River flood, showing values at building locations with columns for: [A] depth; [B] difference (simulated – trusted); and [C] linear correlation of simulated- (y-axis) against trusted-depths (x-axis). Values exceeding 10 m are hidden from row 2 panels for clarity. Common statistical metrics for the plotted data are shown within each plot along with the 'Depth Grid Code' from Table 1. Coloration is applied for convenience when cross-comparing with other figures.**

The somewhat normally distributed depths of the trusted dataset are shown in Fig. 4 Panel A1 with a mean of 0.91 m and a max of 6.01 m. In contrast to this, Fig. 4 Panel A2 and A3 show a more exponential shape for the simulated results. While FwDET-QGIS does not have a built-in zero-filter, the 181 'zero depth' values were removed prior to this performance analysis





as previously noted. Similarly, the ten FwDET-QGIS simulations with depth values exceeding 10 m are hidden on the figure

(these outliers are discussed further below). In contrast to this, the maximum depth estimated by RICorDE is 6.6 m. The disparities in inundation predictions between raster pairs, shown in Fig. 3, is also evident on Fig. 4 in the different sizes of each paired data set, shown as 'count' values on each panel. For example, the 'count=610' shown on Fig. 4 Panel C2 is equivalent to the height of the green bar shown on Fig. 3 'SJ_FQ'. The aforementioned 'dry bias' of RICorDE is also replicated in the depth values shown on Fig. 4 Panel B3.

Fig. 5 presents comparable plots for the 2017 Rivière Des Prairies flood showing similar performance. Interestingly, the shape of the trusted depth values histogram for this trial (Fig. 5 Panel A1) differs from that in the Saint John trial (Fig. 4 Panel A1). This could be a result of differences in topography, development patterns, flood behavior, levee performance, or the hydraulic modelling methods used (by others). While insufficient information was available to evaluate this further, these disparities point to the importance of incorporating multiple heterogeneous trials when evaluating tools like RICorDE.

For the Des Prairies trial, Fig. 5 again shows RICorDE outperforming FwDET-QGIS; with even the mean difference metric being more favorable by 5 cm. However, the comparable and relatively favorable performance of both tools in predicting depths should be weighed against the poor inundation performance shown in Fig. 3 for this trial. This suggests the FiC polygon used in this trial led to many false predictions; however, where the FiC polygon was accurate, RICorDE yielded better depth estimates.

All the performance metrics discussed above are sensitive to the treatment of zero values and paired values (with one dry or missing value). The treatment used here (and described above) was selected to provide broad and clear metrics; however, alternate treatments would also be reasonable. For example, the difference analysis could have discarded any paired values with a missing value, rather than setting the missing value to zero. Many of these alternate metrics were explored by the study-team — all yielded similarly favorable results for the performance of RICorDE.



**Figure 5: Performance metrics for the 2017 Rivière Des Prairies flood as in Figure 4.**

To further investigate the performance of the two tools, Fig. 6 provides an overview and two comparable maps of the 2018 Saint John River study event. This shows a portion of the river where the FiC polygon erroneously identified inundation up the riverbank to an elevation of 60 m (lower left of Fig. 6 Panel A and B). FwDET-QGIS interpolated these shoreline values directly onto the interior, until values propagated from the opposing bank were encountered by the routine (Fig. 6 Panel B black arrows) yielding depth errors in excess of 20 m for assets that should have been dry (according to the validation data).





Contrary to this, RICorDE's hydro-correction clipped this inundation to achieve shoreline elevations between 7 and 10 m before interpolating these onto the interior. This mechanism contributed to the overall more accurate predictions of RICorDE

discussed above.

**Figure 6: Gridded depths for the Saint John River, May 2018 flood event showing: [C] trusted depth estimates for the full study area; [A] RICorDE simulations minus trusted depth results with hydraulically corrected inundation; and [B] FwDET-QGIS minus trusted. All sample points with real depth values are labelled.**



## 4 Conclusions


This study developed the Rolling HAND Inundation Corrected Depth Estimator (RICorDE) Tool for predicting depths from approximate inundation polygons. Similar to previous tools, like FwDET-QGIS, RICorDE provides an efficient method that does not assume a flat-water surface and leverages remote sensing data and does not require hydraulic modelling expertise or difficult to obtain calibration data (e.g., bathymetry, high-water marks). Unlike previous tools, RICorDE incorporates some

error correction of approximate inundation polygons and is structured around a HAND sub-model to facilitate more realistic water surfaces. These enhancements come at the cost of algorithm complexity and longer runtimes. This work enhances the utility of satellite derived data for studying flood events; thereby improving society's ability to plan and prepare for flood disasters.

To test the performance of RICorDE, two recent flood events were examined. Depth estimates were generated for these events

from public data and approximate inundation polygons from the Floods in Canada project (FiC) using RICorDE and the popular FwDET-QGIS tool, before comparing against trusted depth grids. The presented results suggest the novel RICorDE's algorithm outperforms FwDET-QGIS in depth predictions for the two study floods investigated with RICorDE having a RMSE of 0.790 and 0.509 m for the two trials. Inundation performance was mixed, with FwDET-QGIS having slightly fewer 'under' predictions (tool predicts the asset is dry in error) while RICorDE had far fewer 'over' predictions (tool predicts the asset is

wet in error), suggesting a slight 'dry bias' for RICorDE. This demonstrates that both algorithms remain limited by inaccuracies in satellite-derived polygons like FiC, with one trial predicting less than half of asset inundations accurately. Future work should consider improving the underlying satellite derived inundations, improving the run times and computational efficiency of RICorDE, testing against different flood hazards (e.g., ice jam flooding), integrating uncertainty quantification, and porting the tool to a more user-friendly environment.

Where RICorDE is used to predict depths at building locations from FiC polygons, the two trials performed here suggest reasonable depth estimates can be obtained for those buildings that were truly inundated; however, the two FiC polygons examined here from 2017 and 2018 were unable to reliably predict this inundation. While the objective of this study was not to test the accuracy of the FiC project, improvements were found between the 2017 and 2018 FiC polygons. This limited observation suggests new satellite missions and internal advancements may bring more useful FiC polygons. With such

inundation polygons, RICorDE provides a more accurate means of studying historical floods remotely and at scale.

**Code Availability**

Code is available by request from the authors.





**Data Availability**

The Saint John River 2018 maximum WSL validation data is hosted by GeoNB (http://www.snb.ca/geonb1/e/DC/floodraahf.asp) under the 'GeoNB Open Data License'. The Rivière des Prairies 2017 maximum WSL trusted data was provided by Brent Edwards (brent.edwards@cmm.qc.ca) of the Communauté métropolitaine de Montréal under license to the project team. Datasets used for the depth tools are publicly available at the locations mentioned in the text.

**Author Contribution**

Motivation for the work stemmed from discussions between the three authors about developing gridded depths from the Floods In Canada project and well-known tools. All authors contributed to collecting and locating the input and trusted data. SB developed the code, performed the simulations and analysis, and prepared the manuscript with contributions from MB and HM.

**Competing interests**

The authors declare that they have no conflict of interest.

**Acknowledgements**

The authors thank Jasmin Boisvert and Reid McLean of the New Brunswick Department of Environment and Local Government for their insight into the 2018 flooding of the Saint John River. We are similarly grateful for the efforts of Brent
Edwards and Nicolas Milot of the Communauté métropolitaine de Montréal for their assistance in obtaining data on the 2017 floods there. Charles Papasodoro at the Canada Centre for Mapping and Earth Observation provided invaluable guidance on integrating HRDEM into the project.

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
