# Peer review of "Gridded Flood Depth Estimates from Satellite Derived Inundations"

_Natural Hazards and Earth System Sciences, 2021_

## Author Response (AR1)

**Gridded Flood Depth Estimates from Satellite Derived Inundations**

**Author's Reply**

Seth Bryant, Heather McGrath, Mathieu Boudreault

We thank the editor and the two reviewers for their insights, comments, and time. The manuscript has been revised following the comments of the reviewers. A detailed point-by-point response to each comment and corresponding action are provided in the following sections. The following additional revisions have been incorporated:

- Minor word choice and grammar refinements

- References to the Whitebox and GRASS projects have been revised to reflect the format requested by the corresponding developers

- The description of the HRDEM project was expanded to emphasize the relation with the hydrodynamic modelling done by others

- Data source information originally documented in the manuscript is now also shown on the Supplements for clarity.

All changes to the text are marked in red or blue on the provided 'track changes' version of the manuscript.

**RC1 Comments**

*This paper describes a DEM-based interpiolation method to estimate flood depth from a flood extent, using the HANDS approach and GIS interpolation operations.*

*I think this paper is generally well written and follows a clear structure, It is technically sound and the method seems fairly straightforward to implement and apply elsewhere.*

*The authors present the methods and validate their results, so, in essence I see nothing wrong with this paper.*

*My only concern is that this reads as (yet) another interpolation method for flood depth from flood extent. In my opinion, the authors need to much better justify why there should be yet another method out there. To make this argument even stronger, the authors should do some kind of senSiTIVITY analysis using different DEMs with varying degrees of accuracy.*

*I do very much appreciate the fact that the authors compare their method to that of Cohen et al but I do not see much difference in performance. Is this correct? This should be explained better.*

*Other than the comments above, I think this is a valid application paper that could be of interest to the community.*

Thank you for your review and comments.

We agree that our method provides only marginal improvement over FwDET when considering the performance metrics focused on total accuracy (RMSE, difference mean, and r-value). However, RICorDE substantially improves the handling of outliers and the generation of more realistic looking depth maps. See for example Figure 6 panel B, where FwDET returned depths nearing 20m for assets that the trusted simulation and RICorDE determined to be dry. An additional performance metric 'difference > 2m (count)' has been added to Table 1 to better articulate RICorDE's advantage in handling outlier samples with egregious errors. Fig. S3 and Fig. S4 have been added to better demonstrate the improvement in visual realism of RICorDE's depth rasters. The following sentence has also been added to Section 4 'Conclusions':

RICorDE substantially outperformed FwDET-QGIS in the treatment of outliers, returning an order-of-magnitude fewer predictions with errors exceeding 2m for one trial.

As for pursuing a sensitivity analysis, we agree this would provide valuable information, especially for potential users desiring to extract depths from satellite derived inundations. However, the aim of our study was only to show an improvement over existing tools, like FwDET. For this, parameters like DEM resolution are less relevant as they would similarly affect the output of both tools, leading to a similar improvement.

**RC2 Comments**

*The presented floodwater depth calculation methodology, RICorDE, is innovative in its coupling of HAND and "raw" elevation data to produce more hydraulically robust results. The manuscript is well written and the authors did an overall good job at explaining the new elements in the workflow. This paper can be of great interest to the community. I have a few concerns:*

*1. The authors opted not to share their code and tool in an open repository - this is their right but disappointing, especially considering that they developed this tool based on open-source resources (primarily FwDET). The tool also seems to be specific to Canadian data sources, maybe a more generic version can be shared. This is not critical for the paper's publication but will considerably increase its impact in my opinion.*

Thank you for your review and comments.

The intention is to have the tool be open source once we can resolve some contractual issues.

The tool is inspired by FwDET but does not rely on any source code from FwDET. However, many other opensource libraries are used (GRASS, QGIS, WBT, etc.). As mentioned on line 114, the tool does include scripts to pre-process from Canadian data sources, but alternate data sources could easily be provided by users.

*2. The evaluation of the tool is based solely on remote sensing-derived flood maps. This choice is understandable but as the paper shows it is hard to isolate the source of the error in the model prediction. The evaluation presented is of great value but the authors can quite easily use the hydraulic-model inundation extent as input, similar to what others have done. The authors justify their choice but it, nonetheless, leads to uncertainty of how much the new methodology is an improvement over FwDET or a result of "improving" the remote sensing errors by shrinking the flooding domain. The reader will benefit from knowing the answer.*

The motivation for our tool is to develop depths from remote sensing data, therefore we used these as inputs to the tool. Others have used inundation from hydraulic model outputs – presumably as an intermediate step towards eventually working with remote sensing data. We provided the same satellite derived inputs to both RICorDE and FwDET and compared the results. The first phase of RICorDE develops a hydraulically derived inundation, removing egregious errors from the flooded domain. This algorithm is a part of RICorDE, and we therefore do not separate it for a comparison against FwDET.

*3. The authors all but ignored the issue of runtime. They mention "longer runtime" in line 336 but offer no further details. This is quite an important aspect for depth calculation from remote sensing as these are often used for flood response and large-scale applications. The authors should report their model and FwDET runtime for their case studies. This can be most useful for future users and developments.*

Our tool is optimized for accuracy – not runtime. This may make it more useful for flood vulnerability research than for disaster response; however, it could still be used in some contexts for disaster response. Run times for the four case-tool combinations have been added to Table 1. For the next version of RICorDE, we will focus on improving these run times (e.g., parallelizing), as mentioned on line 362.

> *4. There are no floodwater depth maps presented with the exception of a very small insert and the "trusted" data. This is a major emission. As the authors discuss, floodwater depth estimations often include sharp transitions (strips) in the map. RICorDE primary premise is in its innovative treatment with boundary cells which has the potential of alleviating this problem. Yet, this is neither presented nor discussed in the manuscript. Reducing unrealistic artifacts in the depth map is important for improving its accuracy and since practitioners are much less likely to trust products that include clear errors.*

Fig. S3 and Fig. S4 have been added to show the resulting depth raster outputs of both tools for both study areas.

> *5. The authors need to provide more information about the "trusted" products. Which models were used, what was the native resolution, is the DEM used here is the same as for the simulations, is the remote sensing product capture the same day/conditions as the hydraulic simulation.*

Our objective was to demonstrate RICorDE provides some improvement over FwDET, not to quantify absolute accuracy to a real flood event. Therefore, we thought the nuances of the trusted grids less relevant. Regardless, Section 2.2.1 has been expanded to better document the hydraulic modelling provided to us by CMM and PoNB.

*Minor comments:*

*Lines 90-91: this sentence is technically true but misleading as FwDET average errors were much smaller ("...an average difference of 0.18 and 0.31 m for the coastal (using a 1 m DEM) and riverine (using a 10 m DEM) case studies, respectively.")Table 1: add units to the header of relevant columns*

The average difference values from Cohen et al. you mention have been added and the sentence now reads (addition italicized):

> Cohen et al. (2019) tested this tool against hydrodynamic results for two flood-prone regions in the U.S. using the hydrodynamic inundation for the tool input (rather than satellite derived inundations) and found *average errors of 0.18 and 0.31 m, with some* errors exceeding 1.5 m.

Units have been added to Table 1 (RMSE has been changed to cm to align with other publications)

---

## Author Response (AR2)

**Gridded Flood Depth Estimates from Satellite Derived Inundations**

**Author's Reply v2**

Seth Bryant, Heather McGrath, Mathieu Boudreault

We thank the editor and the reviewer for their insights, comments, and time. The manuscript has been revised following the comments received. A detailed point-by-point response to each comment and corresponding action are provided in the following sections. Where necessary for context, previous comments and responses have been included in grey.

All changes to the text are marked in red on the provided 'track changes' version of the manuscript.

**Editor's Comments**

> *In your response to R1 you state that "As for pursuing a sensitivity analysis, we agree this would provide valuable information, especially for potential users desiring to extract depths from satellite derived inundations." I would suggest to add something about the value of this to the discussion..*

The following sentence has been added to the Conclusions: "Additionally, a sensitivity analysis of the key parameters and input data characteristics (e.g., resolution) would provide useful information for those planning to use tools like RICorDE."

**RC2 Comments**

> *The presented floodwater depth calculation methodology, RICorDE, is innovative in its coupling of HAND and "raw" elevation data to produce more hydraulically robust results. The manuscript is well written and the authors did an overall good job at explaining the new elements in the workflow. This paper can be of great interest to the community. I have a few concerns:*
>
> *1. The authors opted not to share their code and tool in an open repository - this is their right but disappointing, especially considering that they developed this tool based on open-source resources (primarily FwDET). The tool also seems to be specific to Canadian data sources, maybe a more generic version can be shared. This is not critical for the paper's publication but will considerably increase its impact in my opinion.*

Thank you for your review and comments.

The intention is to have the tool be open source once we can resolve some contractual issues.

The tool is inspired by FwDET but does not rely on any source code from FwDET. However, many other opensource libraries are used (GRASS, QGIS, WBT, etc.). As mentioned on line 114, the tool does include scripts to pre-process from Canadian data sources, but alternate data sources could easily be provided by users.
* * *
*RC2 Comment #1 - consider informing the readers about that*
* * *
The 'Code Availability' section has been revised as follows: "Please contact the authors or visit the project page (https://github.com/cefect/RICorDE_pub) to obtain a copy of RICorDE. Future releases of RICorDE are planned to be open source."
* * *
*2. The evaluation of the tool is based solely on remote sensing-derived flood maps. This choice is understandable but as the paper shows it is hard to isolate the source of the error in the model prediction. The evaluation presented is of great value but the authors can quite easily use the hydraulic-model inundation extent as input, similar to what others have done. The authors justify their choice but it, nonetheless, leads to uncertainty of how much the new methodology is an improvement over FwDET or a result of "improving" the remote sensing errors by shrinking the flooding domain. The reader will benefit from knowing the answer.*
* * *
The motivation for our tool is to develop depths from remote sensing data, therefore we used these as inputs to the tool. Others have used inundation from hydraulic model outputs – presumably as an intermediate step towards eventually working with remote sensing data. We provided the same satellite derived inputs to both RICorDE and FwDET and compared the results. The first phase of RICorDE develops a hydraulically derived inundation, removing egregious errors from the flooded domain. This algorithm is a part of RICorDE, and we therefore do not separate it for a comparison against FwDET.
* * *
*RC2 Comment #2 - at least shortly discuss this issue*
* * *
Line 99 states "[…] no study with a fully automated method reported accuracy against satellite derived inundations." The additional sentences have been added to the "Tool Inputs" section: "This differs from many previous studies (Raney and Cohen, 2019; Cohen et al., 2018; Scorzini et al., 2018) that employ inundations from hydrodynamic models in their performance evaluations. For this study, satellite derived inundations of real floods are used in the evaluation to better reflect the intended application of RICorDE and the stated objectives of the tool.  However, this makes it difficult to directly compare the results of our evaluation with the evaluations of others."

*3. The authors all but ignored the issue of runtime. They mention "longer runtime" in line 336 but offer no further details. This is quite an important aspect for depth calculation from remote sensing as these are often used for flood response and large-scale applications. The authors should report their model and FwDET runtime for their case studies. This can be most useful for future users and developments.*

Our tool is optimized for accuracy – not runtime. This may make it more useful for flood vulnerability research than for disaster response; however, it could still be used in some contexts for disaster response. Run times for the four case-tool combinations have been added to Table 1. For the next version of RICorDE, we will focus on improving these run times (e.g., parallelizing), as mentioned on line 362.

*[Referee 2]: RC2 Comment #3 - add a short discussion of the new runtime results.*

*[EDITOR]: Note that with regards to the question of run-time, my understanding is that this is now included in Table 1 already.*

Run time values were previously added to Table 1. In addition, the following sentence has now been added to the "Results and Discussion" section: "These runtimes show that RICorDE, which was designed for accuracy not speed, is substantially slower than FwDET-QGIS."

*4. There are no floodwater depth maps presented with the exception of a very small insert and the "trusted" data. This is a major emission. As the authors discuss, floodwater depth estimations often include sharp transitions (strips) in the map. RICorDE primary premise is in its innovative treatment with boundary cells which has the potential of alleviating this problem. Yet, this is neither presented nor discussed in the manuscript. Reducing unrealistic artifacts in the depth map is important for improving its accuracy and since practitioners are much less likely to trust products that include clear errors.*

Fig. S3 and Fig. S4 have been added to show the resulting depth raster outputs of both tools for both study areas.

*RC2 Comment #4 - the new figures are a very powerful demonstration of the new tool's advantage. Strongly consider adding at least one of them to the manuscript.*

A detail map comparing the three gridded depth results has been added as the new Figure 3.